# A Self-Supervised Tree-Structured Framework for Fine-Grained Classification

**Qihang Cai** , **Lei Niu** * , **Xibin Shang and Heng Ding**

Central China Normal University Wollongong Joint Institute, Faculty of Artificial Intelligence in Education, Central China Normal University, Wuhan 430079, China
* Correspondence: lniu@ccnu.edu.cn

**Abstract:** In computer vision, fine-grained classification has become an important issue in recognizing objects with slight visual differences. Usually, it is challenging to generate good performance when solving fine-grained classification problems using traditional convolutional neural networks. To improve the accuracy and training time of convolutional neural networks in solving fine-grained classification problems, this paper proposes a tree-structured framework by eliminating the effect of differences between clusters. The contributions of the proposed method include the following three aspects: (1) a self-supervised method that automatically creates a classification tree, eliminating the need for manual labeling; (2) a machine-learning matcher which determines the cluster to which an item belongs, minimizing the impact of inter-cluster variations on classification; and (3) a pruning criterion which filters the tree-structured classifier, retaining only the models with superior classification performance. The experimental evaluation of the proposed tree-structured framework demonstrates its effectiveness in reducing training time and improving the accuracy of fine-grained classification across various datasets in comparison with conventional convolutional neural network models. Specifically, for the CUB 200 2011, FGVC aircraft, and Stanford car datasets, the proposed method achieves a reduction in training time of 32.91%, 35.87%, and 14.48%, and improves the accuracy of fine-grained classification by 1.17%, 2.01%, and 0.59%, respectively.

**Keywords:** fine-grained classification; tree-structured framework; machine-learning matcher; convolutional neural network





## 1. Introduction

With the continuous development of deep learning algorithms, machines are now able to simulate human visual systems in order to recognize objects. Fine-grained visual recognition has become a critical issue in the area of recognizing objects whose main task is to recognize subclasses from several clusters. The term "cluster" refers to a grouping of objects that share a particular feature, while a "subclass" refers to breeds within the same cluster. Figures 1 and 2 illustrate clusters of planes, respectively. The aircraft in Figure 1 are passenger planes, and the aircraft in Figure 2 are warplanes. Due to clear differences in appearance, it is easy to distinguish between the clusters of warplanes and passenger planes. However, precisely distinguishing between subclasses within the same cluster presents a challenge due to the small visual differences; for example, differentiating Typhoons from other types of warplanes shown in Figure 2 is particularly difficult.

Most contemporary researchers have focused on extracting local discriminative features from images [1], i.e., searching for the most prominent parts of the image. To achieve this, various methods have been developed, such as part-based convolutional neural networks [2], fine-grained three-dimensional networks [3], and feedback-control neural networks [4]. In recent years, convolutional neural networks (CNNs) have gained popularity in solving fine-grained classification problems [5]. To extract features, researchers have proposed innovative techniques such as a recurrent attention convolutional neural network [6] and a DenseNet-based classification framework [7].

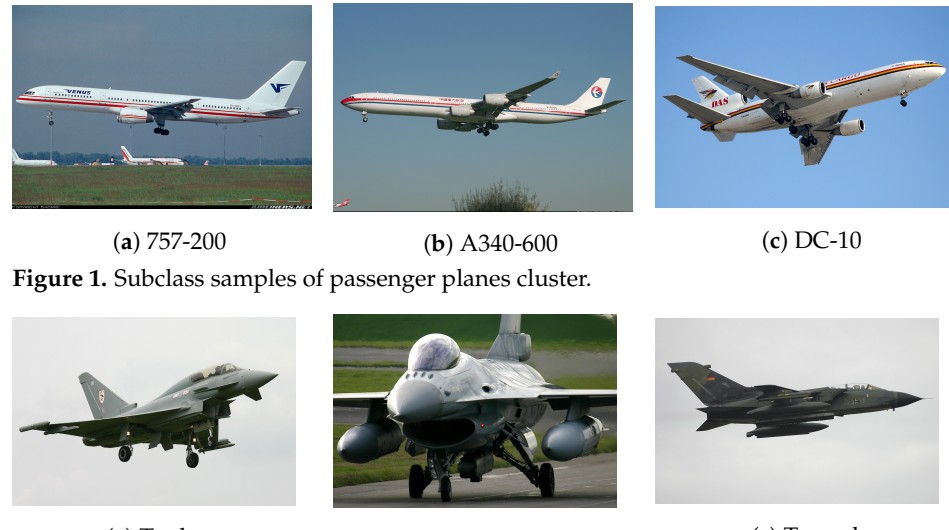

(**a**) 757-200   (**b**) A340-600   (**c**) DC-10

**Figure 1.** Subclass samples of passenger planes cluster.

(**a**) Typhoon   (**b**) F-16AB   (**c**) Tornado

**Figure 2.** Subclass samples of warplanes cluster.

Moreover, solving fine-grained classification problems using traditional CNNs can be challenging. Extracted features often correlate with the image quality and the similarity of samples in the dataset, leading to suboptimal performance. To improve classification accuracy, many researchers focus on removing image noise, such as the background, to enhance image quality [8]. In this paper, we propose a novel framework for fine-grained classification that eliminates the effect of cluster similarity, which is distinct from traditional noise elimination methods. A dataset may contain different clusters, such as warplanes and passenger planes. However, when all samples (subclasses from various clusters) are combined for classification, a CNN classifier must differentiate not only the cluster but also the subclass. This is a complex task because the correlation between different clusters may affect the extracted features, leading to an inaccurate classification. To address this issue, it is essential to identify the belonging cluster for each subclass before the classification. For example, the 757–200 aircraft shown in Figure 1 should be classified into its belonging cluster (i.e., passenger planes) before further classification.

This paper advances a novel tree-structured framework with the objective of enhancing classification accuracy and mitigating training time in comparison to CNNs for fine-grained classification tasks. The proposed framework determines the belonging cluster before classification, which can help to eliminate the effect of the similarity between different clusters. The paper makes several contributions: (1) a self-supervised method that can automatically construct an initial classification tree based on image features, which distinguishes the belonging cluster and eliminates the need for manual labeling; (2) a normal machine-learning matcher which acts as a classifier that utilizes machine-learning methods to assign objects to the belonging clusters, thus minimizing the impact of clusters on the classification process; and (3) a pruning criterion is proposed to filter the classifier with better classification performance, thereby reducing the computational cost. The experimental results demonstrate that the proposed tree-structured classifier improves the accuracy and training time of fine-grained classification compared to traditional CNN classifiers.

The paper is organized as follows. Section 2 provides a discussion of related work. Section 3 introduces a method for the proposed tree-structured classifier. Section 4 discusses the experiments. Section 5 concludes the paper.

## 2. Related Work

Fine-grained image classification is a task that involves identifying subtle visual characteristics of subclasses within a cluster, which generates significant interest among researchers. Several approaches are proposed to address the problem of fine-grained classification. The mainstream approach is "landmark selection," i.e., extracting local dis-

criminative features. For instance, Zhang et al. [9] propose an approach that autonomously identifies informative regions using an object localization module, achieving high classification accuracy without the need for partial manual annotation. Breiki et al. [10] utilize a self-supervised model to automatically detect and annotate informative features in images, leading to improved classification accuracy. Lai et al. [11] segment images into multiple scales to reduce the adverse effect of background information and extract more precise features. In general, the extraction of discriminative features necessitates a substantial amount of computational resources. In contrast, this paper proposes a hierarchical structure to eliminate the influence of different clusters on feature extraction with fewer resources. This section presents a review of three types of related works: (1) feature extraction for entire objects using CNNs, (2) hierarchical classification framework, and (3) hierarchical fine-grained classification.

The extraction of features for entire objects focuses on extracting useful information from the global image by generating a multi-scale comparison of features [12]. CNNs are helpful tools to acquire discriminative features of entire objectives. Zhang et al. [12] use a CNN with proposed multi-max pooling to extract the information of entire objects. Through clustering algorithms, features with high differentiation are selected. Wang et al. [13] apply DenseNet-121, 169, and 201 to extract the features of multiple sclerosis. However, due to the similarity of subclasses, there is still room to improve the accuracy of fine-grained classification. Hu et al. [14] add a spatially weighted pooling layer of CNNs for better feature extraction. These methods focus on extracting discriminative features rather than dealing with the effect of the similarity between different clusters on feature extraction.

The hierarchical classification framework has seen a surge of work in recent years, e.g., speech recognition, product selection, and text classification. The hierarchical classification framework mainly has two effects: (1) the extraction of useful features, e.g., hierarchical neural networks [15], and hierarchical graph neural networks [16], and (2) the simplification of multiclassification problems into a few small classification problems, which is the basic idea of this paper. Daume et al. [17] automatically build a tree hierarchy based on the recall of test samples. Although the classifier based on the recall hierarchy improves the speed of classification, the accuracy slightly decreases. Furthermore, more approaches to building the hierarchical classifier are based on the features of the samples [18–20]. Morin et al. [18] construct a binary hierarchical classifier based on prior knowledge, which is extracted from the semantic feature by WordNet. The study only discusses the effect of the binary hierarchical classifier in speech recognition but does not explain the condition in which the tree hierarchy stops growing and discusses the outcome of the tree hierarchy with other numbers of child nodes. Based on extracted features, Yu et al. [20] construct a hierarchical classifier using machine-learning clustering, which can narrow down the candidate set from an enormous output space and find the most relevant items. However, the authors do not discuss the performance of the hierarchical classifier constructed with different machine-learning approaches. In general, most studies do not discuss the effect of autonomous and self-supervised hierarchical frameworks in fine-grained classification.

Hierarchical fine-grained classification improves the accuracy of fine-grained classification by building hierarchies for classification. Based on object features, the idea of hierarchical classification classifies the objects into different clusters, and each cluster has high similarities. As the number of levels increases, the similarity in each cluster increases [21]. Goel et al. [22] create a binary tree hierarchy using a max-margin clustering strategy. A metric learning strategy is applied to learn the difference between objects in metric space. In [22], a binary tree hierarchy is built based on the visual features rather than the features extracted by CNNs, i.e., manual labeling is required. Similarly, Bameri et al. [23] propose a hierarchical method for classification based on the phylogenetic information of birds, which means that additional information is needed when processing the dataset. In the above approaches [22,23], manual calibration is required to build the tree hierarchy, which requires various efforts.

In summary, the current literature focuses on extracting discriminative features that require significant computational costs when extracting features of entire objects. However, Tanno et al. show that the tree hierarchy requires fewer costs in their work [15]. Meanwhile, researchers have investigated the effect of hierarchical structures in neural networks extensively. However, little work has been carried out on autonomous and self-supervised hierarchical frameworks in fine-grained classification. Furthermore, manual marking is usually the basis for fine-grained hierarchical classification, which requires high labor costs, and there is little research on self-supervised methods. Therefore, it is worth constructing accurate, fine-grained, self-supervised hierarchical classifications with entire features to reduce the effect of the similarity between different clusters.

## 3. Proposed Tree-Structured Framework for Fine-Grained Classification

In the pursuit of developing a framework for fine-grained classification, we draw inspiration from the conventional hierarchical classification approach, which involves constructing a tree hierarchy of objects based on their features. The effectiveness and accuracy of traditional hierarchical classification heavily depend on constructing an accurate and well-structured tree hierarchy. This process typically involves two stages: (i) utilizing relevant background knowledge to establish a reasonable tree hierarchy and (ii) manually labeling images with the appropriate cluster. However, the manually constructed tree hierarchy may not be applicable to general fine-grained classification problems, as it may be challenging to determine the optimal tree structure without prior knowledge, and labeling objects is a time-consuming task. To overcome these limitations, this study proposes an automated and modular framework for solving general fine-grained classification problems, as depicted in Figure 3. The framework entails two phases: initial classification tree construction and tree-structured classifier building.

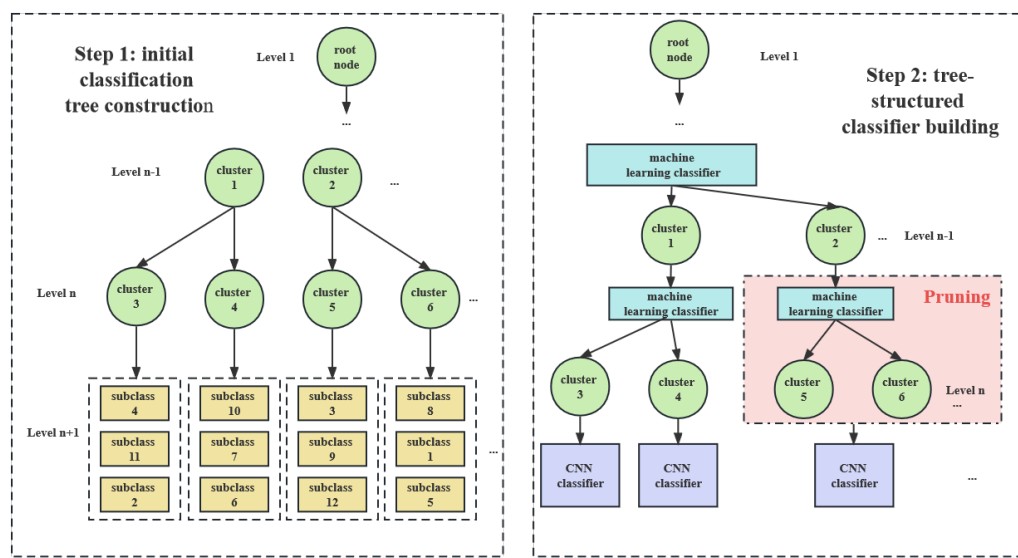

**Figure 3.** The process of constructing a tree-structured classifier.

1.  **Initial classification tree construction**: This step applies a recursive method to construct an initial classification tree through a clustering algorithm. This helps to group similar subclasses together, achieving the effect of automatic hierarchy building. As shown in Figure 3, the proposed approach classifies subclasses into different clusters based on their similarity in Level $n$. Then, the clusters of Level $n$ are classified into Level $n - 1$ clusters.
2.  **Tree-structured classifier building**: In this step, a machine-learning matcher is developed to predict the belonging cluster. Then, a pruning criterion is applied to remove useless clusters from the tree hierarchy, resulting in a classifier with a better classifica-

tion performance. As shown in Figure 3, the red box represents the useless clusters that have been pruned. Finally, within the leaf cluster, a CNN model is used to predict the subclass.

### 3.1. Initial Classification Tree Construction

The initial classification tree constructed based on a clustering algorithm has several advantages. First, the clusters are constructed based on the similarity of subclasses, meaning that all subclasses in each cluster share certain common properties, which is an important factor that can help the machine-learning matcher to accurately predict the belonging cluster. Furthermore, the size of the output space in cluster $i$ is $n_i$, which is much smaller than the size of the overall output space $o$ ($n_i << o$) [20]. This greatly reduces the cost of computation and improves the efficiency of the classification process. This section introduces the process of constructing a well-structured initial classification tree.

As the basis of initial classification tree construction, features of images can be extracted with pre-trained models [24], fine-tuned models [25], and so on. Furthermore, CNNs, e.g., ResNet [26], are one of the mainstream tools for extracting image features. In this paper, a fine-tuned CNN model is utilized to extract features to represent the images.

Once the image features are extracted, an initial classification tree can be constructed through a clustering algorithm such as K-means++ [27]. The purpose of the clustering algorithm is to classify features into the belonging cluster at each level. Given the set of image features $F_i^{n-1} = \{x_i^{n-1,\,1}, x_i^{n-1,\,2}, \ldots, x_i^{n-1,\,n_i}\}$ belonging to a cluster $i$ at level $n-1$ ($C_{i,n-1}$), we aim to determine the belonging cluster $C = \{C_1^n, C_2^n, \ldots, C_k^n\}$ at Level $n$ for $F_i^{n-1}$. Below, an objective function of the clustering algorithm is shown.

$$\{C_l^n \mid l = 1, 2, \ldots, k\}$$
$$\text{s.t.} \quad \begin{cases} C_{l'}^n \bigcap_{l' \neq l} C_l^n = \varnothing \\ F_i^{n-1} = \bigcup_{l=1}^{k} C_l^n \end{cases} \tag{1}$$

Figure 4 illustrates an example of constructing one part of the initial classification tree, where Feature $F_i^{n-1}$ is assigned to Cluster $C_i^{n-1}$. Subsequently, the structure of Level $n$ is constructed by applying the clustering algorithm.

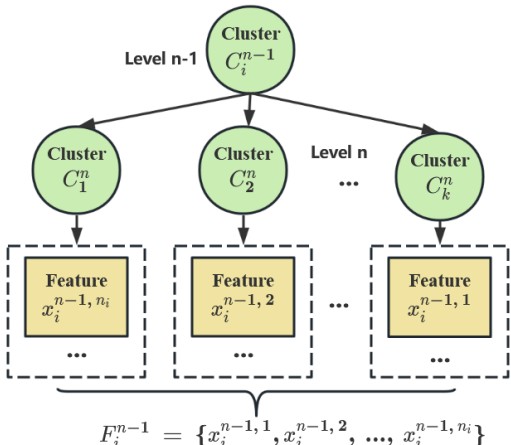

**Figure 4.** A part of the initial classification tree constructed with the clustering algorithm.

Now, the belonging cluster for each image feature has been obtained. However, usually, each subclass contains various images in the data. Hence, it is necessary to determine the belonging cluster for each subclass. Algorithm 1 introduces the process of determining the belonging classes through the voting strategy.

---

**Algorithm 1** Determining the belonging classes for the subclasses.

---

**Input:** Extracted features from each image in the training set (denoted as $F$); Number of the subclasses (denoted as $n_{sub}$); Number of clusters split from the cluster in the previous level (denoted as $n_{chd}$)

**Output:** belonging cluster for each subclass

1: clustering algorithm applied to segment $F$ in $n_{chd}$ clusters
2: **for** $i = 1$ to $n_{sub}$
3:   **for** $j = 1$ to $n_{chd}$
4:     count the number of image features in each cluster $C_k$, where $k \in [1, n_{chd}]$ (voting)
5:     the highest number of votes is selected as the belonging cluster
6:   **end for**
7: **end for**

---

Algorithm 1 determines the belonging clusters for subclasses according to image features. First, the inputs are specified as the extracted features from each image, the total number of the subclasses, and desired number of clusters split from the cluster in the previous level. Then, a clustering algorithm is applied to classify the image features (Line 1). The number of image features in each cluster is calculated for each subclass (Line 4). Following the principle of majority rule, the belonging cluster for each subclass is determined based on the highest number of votes (Line 5). The algorithm stops until it determines the belonging cluster for all subclasses.

Once belonging clusters of the subclasses have been determined, we can dynamically construct the initial classification tree by performing a preorder traversal, which allows us to decide the number of levels in the classification tree and the number of clusters split from the cluster in the previous level. The construction process is shown in Algorithm 2.

---

**Algorithm 2** Construction of the initial classification tree.

---

**Input:** Expected number of levels in the classification tree (denoted as $n_l$); Number of clusters split from the cluster in the previous level (denoted as $n_{chd}$); Extracted features for each image (denoted as $F$);

**Output:** an initial classification tree (denoted as $t_{int}$)

1: **function** CreateClassificationTree (feature, cluster)
2:   **while** levels of classification tree $< n_l$ **do**
3:     determine the belonging cluster for the subclass (Algorithm 1)
4:     **for** $i = 1$ to $n_{chd}$ **do**
5:       cluster $\leftarrow$ cluster $i$;
6:       feature $\leftarrow$ feature of cluster $i$ in $F$;
7:       CreateClassificationTree (feature, node)
8:     **end for**
9:   **end while**
10: **end function**

---

In Algorithm 2, the initial classification tree is constructed recursively based on the image features. First, the inputs are specified as the expected number of levels in the classification tree, the number of clusters split from the cluster in the previous level, and extracted features for each image. A clustering algorithm is used to segment clusters of image features, and majority voting is used to determine a belonging cluster for each subclass (Line 3). Construction of initial classification trees is built using preorder traversal with recursion (Lines 2–9). The cluster is set to the current cluster $i$ (Line 5), and the feature is updated to the values in the current cluster $i$ (Line 6). The recursion ends and outputs the initial classification tree only when the expected number of levels and the desired number of clusters resulting from the split in the previous level have been reached.

### 3.2. Tree-Structured Classifier Building

In the previous section, an initial classification tree was constructed, which could classify each subclass into the belonging class. This section focuses on building a tree-structured classifier with higher accuracy.

As one of the crucial components in the tree-structured classifier, machine-learning matchers are applied to match image features to the belonging cluster in the different levels of the classification tree. Because the final subclass is limited to the cluster returned by the matcher, constructing an effective matcher function is crucial to the accuracy of the image classification. Typically, the dimensionality of the image feature $F$ is higher than that of the cluster labels, which means that it is essential to construct a general matcher function, denoted as $g(\cdot)$, that can find associations between image features $x_i^j$ and the corresponding clusters $C_i^j$ in Level $j$, as shown in Equation (2). $C_i^j$ is the cluster with the highest probability in $C_k^j$ as judged by $g(\cdot)$.

$$C_i^j = \underset{x_i^j \in F \ C_k^j \in C_j}{\arg\max} \ g(x_i^j, C_k^j) \tag{2}$$

Figure 5 illustrates an example of matching an image feature to its belonging cluster. The input image feature is passed through the tree, and a machine-learning matcher determines the belonging cluster (denoted as the red node).

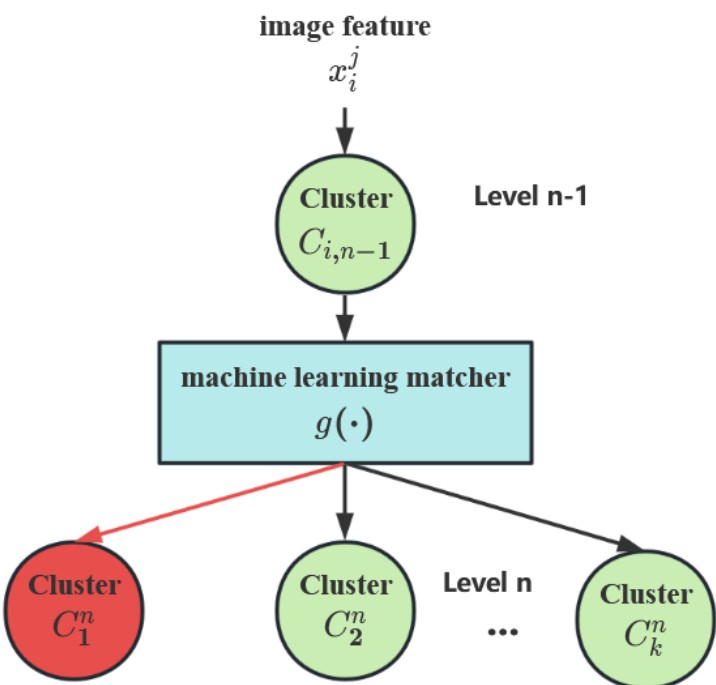

**Figure 5.** Matching an image feature to its belonging cluster.

Now, it is able to predict the belonging cluster for each image. Then, it is necessary to determine subclasses for images with higher accuracy, i.e., the primary objective of this paper is to develop tree classifiers that can achieve higher accuracy. The theory of recall trees [17] shares a similar goal, aiming to ensure that each level of the tree-structured classifier enhances classification accuracy. According to Daume et al.'s findings [17], the accuracy of a tree classifier increases up to a certain threshold as the number of levels increases. However, further increasing the tree's levels beyond this threshold can actually decrease accuracy. As a result, it is crucial to eliminate clusters that have a negative impact on the tree-structured classifier's accuracy.

In this section, we present an algorithm (Algorithm 3) that can create a tree-structured classifier with improved accuracy by eliminating invalid clusters. Prior to that, it briefly introduces a condition for the tree hierarchy to stop growing, i.e., the pruning criterion.

**Pruning criterion**: If the accuracy of a cluster is greater than or equal to the total accuracy of all child clusters, as illustrated in Equation (3), then the child clusters are considered invalid and are deleted from the tree-structured classifier.

$$\exists\, C_i^{j-1}, C_s^j : f(C_i^{j-1}) \geq \alpha \sum_{s=1}^{k} \frac{n_s^j}{n_i^{j-1}} f(C_s^j) \tag{3}$$

In Equation (3), $C_i^{j-1}$ refers to Cluster $i$ in Level $j-1$, $C_s^j$ refers to child cluster $s$ in Level $j$, $f(\cdot)$ refers to the function used to calculate the classification accuracy within a cluster, $k$ refers to the number of child clusters, $\alpha$ refers to the accuracy on prediction of the cluster to child clusters by the machine-learning matcher, $n_s^j$ refers to the number of subclasses in cluster $C_s^j$, and $n_i^{j-1}$ refers to the number of subclasses in cluster $C_i^{j-1}$.

Figure 6 shows an example of a pruned classification tree, which is a binary tree with four levels. The clusters marked in red in Figure 6 are directly deleted based on the pruning criterion, as the accuracy of Cluster 1 is greater than or equal to the accuracy of its child clusters, i.e., total accuracy of Cluster 3 and Cluster 4.

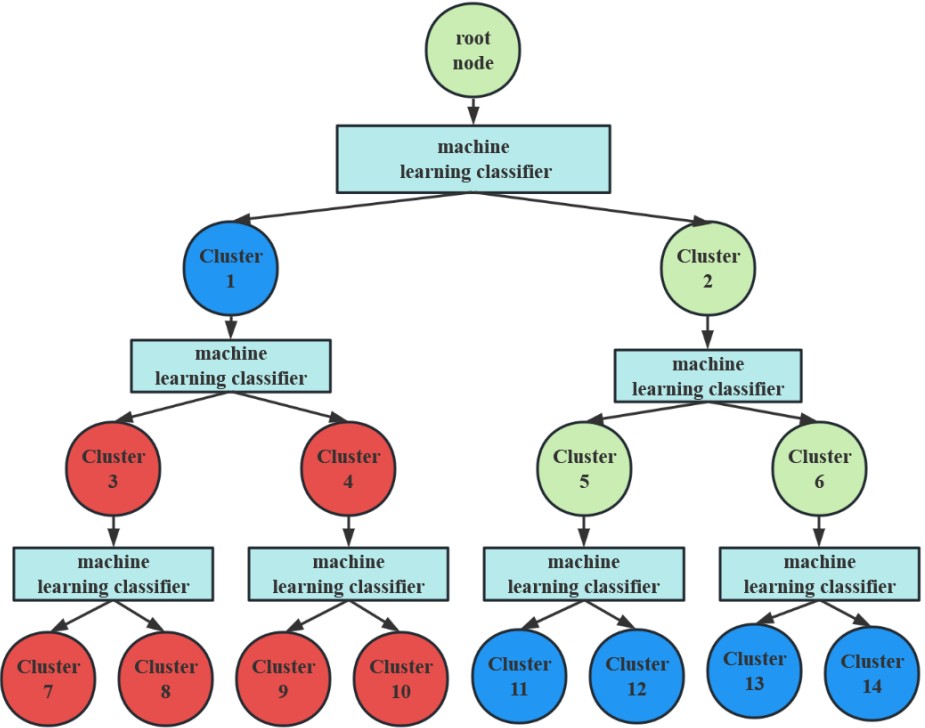

**Figure 6.** An example of classification tree with pruning.

The conditions for stopping the growth of the tree hierarchy have been introduced. Next, Algorithm 3 demonstrates the process of constructing a tree-structured classifier using a hierarchical traversal method to remove unnecessary clusters. Furthermore, CNN classifiers are applied to perform fine-grained classification in each leaf cluster, e.g., clusters marked in blue in Figure 6.

---

**Algorithm 3** Tree-structured classifier building.

---

**Input:** an initial classification tree (denoted as $t_{int}$); machine-learning matcher (denoted as $g(\cdot)$);

**Output:** A tree-structured classifier

  1: **function** CreateTreeStructuredClassifier ($t_{int}, g(\cdot)$)

  2:     Initialize a queue ($q$)

  3:     Put the root cluster of $t_{int}$ into $q$

  4:     **While** ($q$ is not empty) **do**

  5:       Fetch the head node ($n_h$) of $q$

  6:       Calculate the accuracy of $n_h$ ($acc(n_h)$)

  7:       Calculate the comprehensive accuracy of all child clusters of $n_h$ ($acc(n_{chd})$)

  8:       **if** $acc(n_h) < acc(n_{chd})$

  9:         Put all child nodes of $n_h$ into $q$

10:       **end if**

11:     **end while**

12:     apply CNN classifiers for classification in leaf nodes

13: **end function**

---

Algorithm 3 constructs a tree-structured classifier and removes useless clusters through a hierarchical traversal approach. The inputs consist of an initial classification tree and machine-learning matchers. The process starts by initializing an empty queue and placing the root node of the tree in the queue (Lines 1–2). The construction of the tree-structured classifier is built using the hierarchical traversal method (Lines 3–11). During the traversal process, the head node of the queue is fetched (Line 5), and the classification accuracy is calculated (Line 6). Next, Algorithm 3 iterates over and calculates the comprehensive classification accuracy of all child nodes (Lines 7). According to the pruning criterion, it is determined whether the classification accuracy increases with the increase of levels. If the accuracy increases, the child nodes are placed in the queue. Finally, for the leaf nodes in the pruned classification tree, CNN classifiers are applied to perform fine-grained classification in each cluster (Line 12).

This section presents a self-supervised framework for automatically constructing a general tree-structured classifier for fine-grained classification without requiring manual labeling. The process of construction involves classifying each subclass into the belonging cluster based on similarity to eliminate the effect of cluster similarity in feature extraction. Then, a machine-learning classifier is trained to accurately predict the images' belonging cluster using the extracted features. The pruning criterion is proposed to improve the accuracy of the tree-structured classifier by eliminating invalid clusters. Finally, the CNN models are used to identify the belonging subclass of each object in the leaf cluster of the tree-structured classifier.

## 4. Experiment

This section presents four experiments to showcase the performance of the proposed tree-structured framework with diverse structures on different datasets. Experiment aims to explore the impact of the tree-structured classifier's structure on the results and illustrate the necessity of the proposed pruning criterion. Then, the construction of the tree-structured classifier relies on the clustering algorithm and the machine-learning matcher, and the best-performing clustering algorithm and machine-learning matcher are determined through Experiments 2 and 3. Lastly, the purpose of Experiment 4 is to showcase the effectiveness and robustness of the proposed framework, utilizing the superior performance of the k-means++ algorithms and machine-learning matchers identified in Experiments 2 and 3. Specifically, the main objective of Experiment 4 is to investigate whether the proposed framework can enhance accuracy and reduce training time on different datasets in comparison to different traditional CNN models.

### 4.1. Experimental Setting

To evaluate the effectiveness of the proposed approach, this paper assesses the performance of the proposed tree-structured classifier with different structures across different datasets. In this subsection, we provide an overview of the experimental setup from three perspectives: (1) datasets, (2) metric, and (3) parameter settings.

#### 4.1.1. Datasets

The efficacy of the tree-structured classifier is assessed by evaluating its performance on benchmark datasets for fine-grained visual classification of aircraft, birds, and cars. These datasets are FGVC aircraft [28], CUB 200 2011 [29], and Stanford car [30], respectively. Notably, these datasets are established benchmarks for fine-grained aircraft classification. The aircraft dataset consists of 10,000 images and encompasses 100 subclasses, the bird dataset comprises 11,788 images and contains 200 subclasses, and the car dataset comprises 16,191 images and 196 subclasses. Furthermore, the datasets are uniformly segregated into three sets, namely, training, validation, and testing, with a ratio of approximately 1:1:1.

#### 4.1.2. Metric

**Accuracy**: To evaluate the performance of the model, the accuracy of the model is defined in Equation (4).

$$c = \sum_{k=1}^{n_{chd}} \prod_{i=1}^{m} \alpha_i \frac{n_i}{N} \frac{R_a}{R} \tag{4}$$

In Equation (4), $\alpha_i$ refers to the accuracy of matching the cluster to child clusters on Level $i$, $n_{chd}$ refers to the number of clusters split from the cluster in the previous level, $m$ refers to the level of a leaf cluster in a tree-structured classifier, $n_i$ refers to the number of subclasses in cluster $i$, $N$ refers to the number of total subclasses, $R_a$ refers to the number of images accurately classified in cluster $i$, and $R$ refers to the number of images in cluster $i$.

**Training Time**: Given that the training of CNN models is parallel in each layer, the training time of each layer in the tree classifier is determined by the maximum of the training time of the CNN model ($max(t_c^{k,j})$), the training time of the machine-learning matcher ($t_m^{ma}$), and the training time to divide clusters ($t_m^k$). The overall training time ($t$) is defined in Equation (5).

$$t = \sum_{k=1}^{m} (max(t_c^{k,j}) + t_m^{ma} + t_m^k) \text{ where } k \in [1, n_{chd}] \tag{5}$$

In Equation (5), $m$ refers to the level of a leaf cluster in a tree-structured classifier, and $n_{chd}$ refers to the number of clusters split from the cluster in the previous level.

#### 4.1.3. Parameter Settings

Table 1 shows the parameter settings of k-means++ applied in the experiments.

**Table 1.** Parameter settings of k-means++.

| Parameter | Setting Value | Parameter | Setting Value |
|---|---|---|---|
| Maximal number of iterations | 100 | Number of replicates | 1 |
| Start mode | sample | Threshold for change in the cost function | $10^{-4}$ |

Figure 7 illustrates the structure of the CNN model utilized in all experiments. It comprises a CNN model (ResNet, Inception, or MobileNet) and three fully connected layers with either relu or softmax activation functions. Additionally, Table 2 provides a detailed

overview of the parameter settings for the CNN model. Furthermore, Table 3 shows the parameter settings for Random Forest, Naive Bayes, and SVM in the experiments.

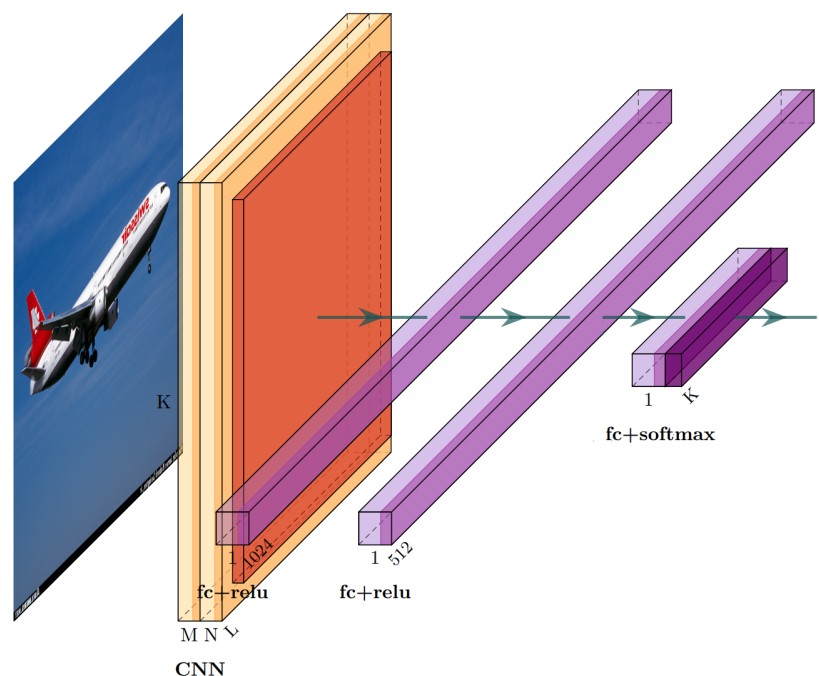

**Figure 7.** The structure of the CNN model.

**Table 2.** Parameter settings for the CNN model.

| Parameter | Setting Value | Parameter | Setting Value |
|---|---|---|---|
| EPOCHS | 80 | optimizer | SGD |
| steps per epoch | Number of subclass/16 | loss | categorical cross-entropy |
| verbose | 1 | metrics | accuracy |

**Table 3.** Parameter settings for machine-learning matchers.

| Machine Learning Matcher | Parameter | Setting Value | Parameter | Setting Value |
|---|---|---|---|---|
| Random forest | number of trees in the forest | 500 | bootstrap sampling for growing trees | True |
| | minimum size of terminal nodes | 1 | maximum number of terminal nodes | NULL |
| Naive Bayes | smallest possible positive number | $e^{-10}$ | frequency-based discretization | False |
| SVM | type | C-classification | coef0 | 1 |
| | cost | 10 | gamma | 0.0009 |
| | probability | Ture | – | – |

## 4.2. Experimental Results

This subsection outlines the training and testing phases based on the initially constructed classification tree. Subsequently, the classification accuracy of various tree-structured classifiers with different structures is assessed.

Training and Testing Phase

**Training Phase**: The training phase encompasses two primary steps, which are illustrated in Figure 8.

1.  **Initial classification tree construction**: Algorithm 2 is employed to construct an initial classification tree, where the K-means++ clustering algorithm is utilized.
2.  **Training to predict class**: Machine-learning matchers are trained for each level of the classification tree to associate images with their corresponding clusters. The tree-structured classifier is built using Algorithm 3, which utilizes the initial classification tree obtained in Step 1 and the trained machine-learning matchers.

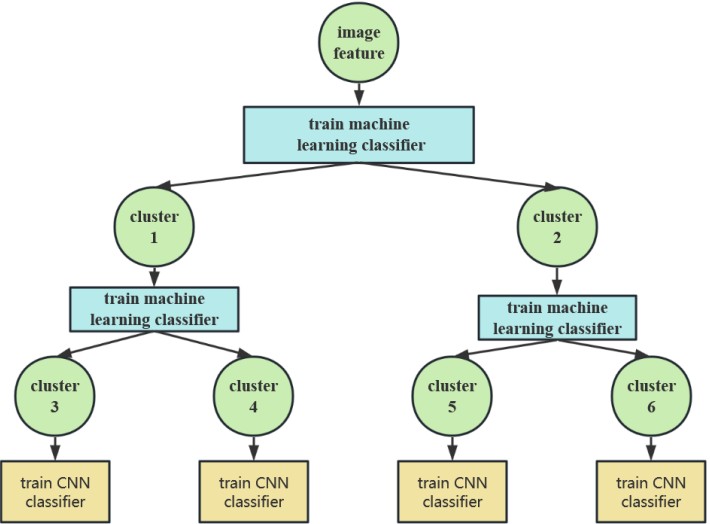

**Figure 8.** The diagram of the training phase of the tree-structured classifier.

**Testing Phase**: The testing phase, based on the trained tree-structured classifier, comprises the following two primary steps.

1.  **Cluster matching**: Based on the extracted image features, a machine-learning matcher is employed to match the belonging cluster, i.e., the leaf cluster.
2.  **Subclass prediction**: The CNN classifier is applied to predict the belonging subclass in the leaf cluster.

### 4.3. Conducted Experiments and Results

To assess the effectiveness of the proposed approach, we executed four experiments and presented their outcomes as follows.

### 4.3.1. Experiment 1

Experiment 1 aims to examine the influence of child clusters on the training time and accuracy of the bird dataset. A tree-structured classifier with varying numbers of child clusters (i.e., 2, 3, 4, and 5) is employed. The accuracy and training time of the classifier are being evaluated to assess the impact of child clusters on the classification performance while holding other variables constant. Specifically, a Support Vector Machine (SVM) with a linear kernel function is used as the machine-learning matcher, ResNet [26] is utilized as the CNN model, and a k-means++ algorithm with Euclidean distance is implemented for clustering. Table 4 presents the impact of the number of child nodes on both classification accuracy and training time. To begin with, in order to showcase the effect of training the proposed model, Figure 9 is provided, which displays partial images of the training and validation loss.

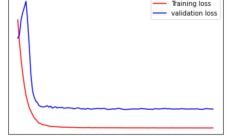 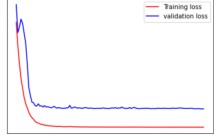 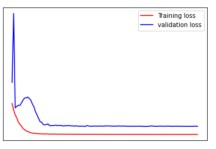 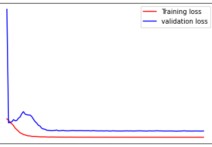

(**a**) Two child clusters　(**b**) Three child clusters　(**c**) Four child clusters　(**d**) Five child clusters

**Figure 9.** Training and validation loss of proposed classifiers with different child clusters.

**Table 4.** Performance of proposed method with different child clusters.

| Number of Child Clusters | Traditional ResNet Classifier Accuracy | Proposed Classifier Accuracy | Traditional ResNet Classifier Training Time | Proposed Classifier Training Time |
|:---:|:---:|:---:|:---:|:---:|
| 2 | | 69.80% | | 1395.08 s |
| 3 | 68.63% | 70.15% | 1480.41 s | 993.25 s |
| 4 | | 69.94% | | 927.64 s |
| 5 | | 68.50% | | 873.20 s |

Table 4 presents the classification accuracy of the bird dataset using the ResNet model without the proposed method, which is 68.63%. However, Table 4 shows that the proposed method enhances the accuracy of classification trees with child clusters 2, 3, 4, and 5 by 1.17%, 1.52%, 1.31%, and −0.13%, respectively. These results suggest that the tree-structured classifier with three child clusters performs better in classification than those with other child clusters. Conversely, the accuracy of classification decreases when the number of child clusters is 4 or 5. Even when the number of child clusters is 5, there is a negative growth in classification accuracy. These results indicate that the accuracy of a tree classifier reaches a certain threshold and then declines as the number of clusters increases. Furthermore, Table 4 displays the training time obtained using ResNet, which amounts to 1480.41 seconds. The present study compares the performance of a proposed method, which consists of 2, 3, 4, and 5 child clusters, with the conventional ResNet architecture in terms of training time. According to the results, the proposed method outperforms the conventional ResNet architecture, with corresponding reductions in training time of 5.76%, 32.91%, 37.34%, and 41.01%, respectively. Furthermore, an increasing trend in the reduction of training time with an increase in the number of child clusters is observed, indicating the effectiveness of the proposed approach.

4.3.2. Experiment 2

To assess the impact of different distance calculation methods in clustering algorithms on the accuracy and training time of tree classifiers on the bird dataset, Experiment 2 evaluates the training time and classification accuracy of clustering algorithms using three different distance calculation methods: Euclidean distance [31], Manhattan distance [31], and correlation distance [31]. While keeping other variables constant, the experiment employs a machine-learning matcher using SVM with a linear kernel function and a CNN model using ResNet. Additionally, the number of child clusters is fixed at three. In Table 5, the impact of utilizing k-means++ with varying distance calculation methods is presented. Initially, to illustrate the impact of the proposed classifier model, Figure 10 illustrates partial figures of the training and validation losses of the images using different distance calculation methods.

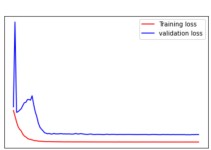 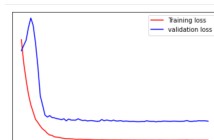 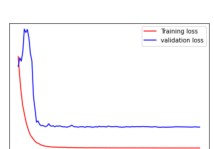

(**a**) Euclidean distance　(**b**) Manhattan distance　(**c**) Correlation distance

**Figure 10.** Training and validation loss of proposed classifiers using different distance calculation.

**Table 5.** Performance of proposed method using different distance calculation methods.

| Distance Calculation Method | Traditional ResNet Classifier Accuracy | Proposed Classifier Accuracy | Traditional ResNet Classifier Training Time | Proposed Classifier Training Time |
|---|---|---|---|---|
| Euclidean distance | 68.63% | 70.15% | 1480.41 s | 993.25 s |
| Manhattan distance | | 70.34% | | 1090.64 s |
| Correlation distance | | 69.03% | | 849.5043 s |

The classification accuracy attained by the traditional ResNet model on the bird dataset is 68.63%, as indicated in Table 5. Furthermore, Table 5 demonstrates that the proposed method enhances the accuracy of classification trees with different distance calculation methods, namely, Euclidean distance, Manhattan distance, and correlation distance, by 1.52%, 1.71 %, and 1.33%, respectively. The results indicate that k-means++ with Manhattan distance can produce better classification results, while correlation distance exhibits the worst accuracy. Generally, k-means++ with different distance calculation methods has improved classification accuracy, which validates the effectiveness of the proposed method. However, the lacking performance of correlation distance might be related to its purpose of calculating the distance, which measures the linear relationship between two variables. On the other hand, Euclidean distance and Manhattan distance are similar since they both measure distance between variables in multi-dimensional space. Hence, during the construction of the initial classification tree, k-means++ with correlation distance places more emphasis on the linear relationship between two variables, which leads to the suboptimal performance of the generated cluster. In terms of training time, the proposed classifier model shows a reduction in training time of 32.91%, 26.33%, and 42.62%, respectively, compared to the conventional Resnet architecture using different distance calculation methods, namely, Euclidean distance, Manhattan distance, and correlation distance. Based on the experimental results, it is concluded that while k-means++ using correlation distance does not result in a significant improvement in accuracy, it requires the least amount of training time. On the other hand, k-means++ with Manhattan distance has the highest accuracy, but it needs the longest training time. The authors suggest that researchers may consider selecting the appropriate k-means++ with different distance calculation methods based on a combination of accuracy and training time.

4.3.3. Experiment 3

To evaluate the impact of various machine-learning matchers on classification accuracy and training time, Experiment 3 tests the performance of three commonly used classifiers for high-dimensional data classification—Bayesian, Random Forest, and SVM—on the bird dataset. This experiment utilizes three classifiers—Naive Bayes, Random Forest, and SVM—with linear, polynomial, radial, and sigmoid kernel functions. While holding the following variables constant, the ResNet architecture is employed as the CNN model, and the k-means++ algorithm with Euclidean distance is utilized for clustering, with a fixed number of three child clusters. Figure 11 delineates not only the structure of Experiment 1 and the quantity of subclusters within each cluster but also emphasizes the machine-learning matchers (identified in blue) that are tested in Experiment 3. Moreover, Table 6 presents the accuracy and training time of the evaluated machine-learning matchers. Table 6 denotes the tested machine-learning matcher as "A–B", where A signifies the number of child clusters of the tree-structured classifier, and B denotes the layer in the tree-structured classifier where the machine-learning matcher is situated.

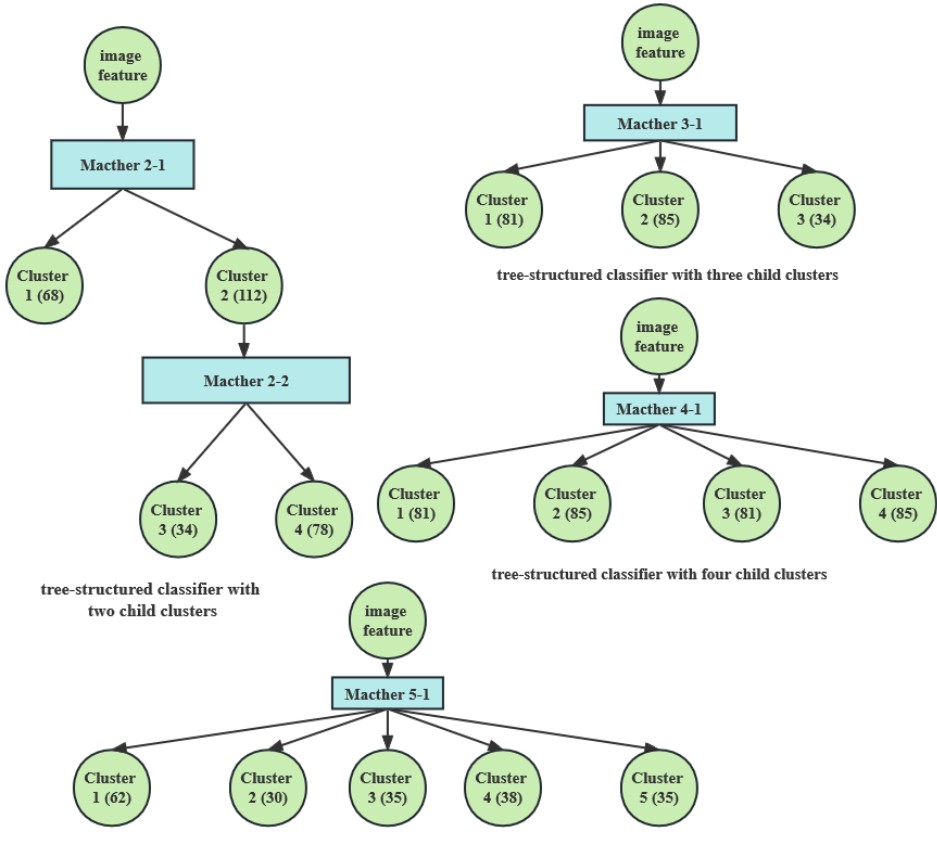

**Figure 11.** Structures of the proposed classifier and tested matchers.

**Table 6.** Performance of tested matchers varies with different machine-learning classifiers.

| Machine-Learning Matcher | Tested Matcher | Accuracy | Training Time | Machine-Learning Matcher | Tested Matcher | Accuracy | Training Time |
|---|---|---|---|---|---|---|---|
| SVM (linear) | 2-1 | 100% | 71.20 s | SVM (radial) | 2-1 | 99.90% | 117.01 s |
| | 2-2 | 100% | 19.41 s | | 2-2 | 98.55% | 30.01 s |
| | 3-1 | 100% | 91.77 s | | 3-1 | 98.55% | 132.89 s |
| | 4-1 | 99.98% | 100.73 s | | 4-1 | 96.85% | 137.91 s |
| | 5-1 | 99.97% | 107.26 s | | 5-1 | 96.68% | 150.48 s |
| SVM (poly) | 2-1 | 100% | 98.86 s | SVM (sigmoid) | 2-1 | 92.29% | 95.61 s |
| | 2-2 | 99.96% | 24.41 s | | 2-2 | 94.18% | 27.14 s |
| | 3-1 | 99.85% | 113.06 s | | 3-1 | 89.70% | 115.24 s |
| | 4-1 | 99.54% | 116.40 s | | 4-1 | 97.67% | 124.69 s |
| | 5-1 | 99.37% | 126.23 s | | 5-1 | 88.63% | 150.90 s |
| Naive Bayes | 2-1 | 91.37% | 0.86 s | Random forest | 2-1 | 98.50% | 209.74 s |
| | 2-2 | 93.14% | 0.64 s | | 2-2 | 97.18% | 101.35 s |
| | 3-1 | 88.56% | 1.13 s | | 3-1 | 96.22% | 225.83 s |
| | 4-1 | 85.64% | 1.31 s | | 4-1 | 95.20% | 233.55 s |
| | 5-1 | 84.22% | 1.51 s | | 5-1 | 94.34% | 234.48 s |

Table 6 indicates that the SVM with linear kernel function is the best classifier, whereas the SVM with sigmoid kernel function and Naive Bayes have poorer accuracy. This is due to Cover's Theorem [32], which suggests that almost all classification problems can be linearly separated in high-dimensional spaces. Hence, the SVM with linear kernel function achieves better accuracy. Additionally, as presented in Table 6, the accuracy of the matcher decreases gradually as the number of nodes increases. For instance, while the accuracy of matcher 2-1 (a machine-learning matcher in a tree-structured classifier with two child clusters in the

first layer, as illustrated in Figure 11) using an SVM with sigmoid kernel function is 92.29%, the accuracy of matcher 5-1 (a machine-learning matcher in a tree-structured classifier with five child clusters in the first layer, as illustrated in Figure 11) drops to 88.63%. It is expected that the difficulty of classification increases with the classification type, resulting in lower accuracy. Likewise, with regard to training time, it can be observed that the training time for all machine-learning classifiers augments as the number of child clusters increases, which implies an increase in the number of categories for classification, such as a binary problem with two child clusters. Notably, the least training time required was for an SVM with a linear kernel function compared to SVM with different kernel functions. Based on accuracy and training time, the authors arrive at the conclusion that SVM with a linear kernel function is the best-performing machine-learning matcher.

### 4.3.4. Experiment 4

To assess the robustness of the proposed method, Experiment 4 measures the performance of various tree-structured classifiers using different CNN models on different datasets, including aircraft [28], birds [29], and cars [30]. The performance of the proposed method is compared with different traditional CNN models, including ResNet [26], Inception [33], and MobileNet [34]. While holding the following variables constant, SVM with a linear kernel function is used as the machine-learning matcher, and the k-means++ algorithm with Euclidean distance is utilized for clustering. Table 7 illustrates the classification accuracy obtained using different structures across diverse datasets. Initially, to demonstrate the effect of training the proposed model on diverse datasets, Figure 12 depicts the parts of training and validation loss figures.

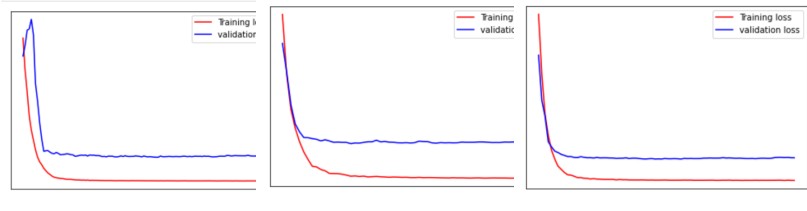

(**a**) Bird dataset     (**b**) Aircraft dataset     (**c**) Car dataset

**Figure 12.** Training and validation loss of proposed classifiers using different datasets.

**Table 7.** Performance of proposed method on various datasets with different child clusters.

| Datasets | Applied Model | Number of Child Clusters | Proposed Classifier Accuracy | Traditional CNN Classifier Accuracy | Proposed Classifier Training Time | Traditional CNN Training Time |
|---|---|---|---|---|---|---|
| Bird | ResNet | 3 | 70.15% | 68.63% | 993.25 s | 1480.41 s |
| Aircraft | MobileNet | 3 | 74.20% | 72.19% | 1962.21 s | 3060.01 s |
| Car | Inception | 2 | 82.83% | 83.42% | 3435.48 s | 4017.29 s |

Table 7 displays the classification accuracy achieved with different structures on various datasets. For instance, using MobileNet to build a tree-structured classifier yields a 2.01% improvement in accuracy. The results indicate that, compared to traditional ResNet, MobileNet, and Inception models, the proposed classifier enhances the classification accuracy of multiple datasets by 1.17%, 2.01%, and 0.59% on the bird, aircraft, and car datasets, respectively. Overall, the proposed tree-structured framework improves classification accuracy in comparison to traditional CNN models. Furthermore, in the airplane dataset, the number of child clusters in the tree-structured classifier is three, resulting in a 2.01% improvement in accuracy. However, in the car dataset, the number of child clusters is two, and the improved accuracy is only 0.59%, which is not particularly significant. This supports the conclusion of **Experiment 1**, where the use of three child clusters exhibits better classification results. Additionally, the result shows that the proposed tree-structured

classifier can considerably reduce the training time compared to the traditional CNN model. This is reflected in a 32.91%, 35.87%, and 14.48% reduction in training time of the bird, aircraft, and car datasets, respectively, compared to the traditional ResNet, MobileNet, and Inception models.

In summary, the aforementioned four experimental outcomes suggest that the proposed approach improves the accuracy and training time of classification in various datasets relative to conventional CNN models. Furthermore, the tree-structured classifier with three clusters yields superior accuracy compared to other configurations, while the training time of the model decreases as the number of child clusters increases. Moreover, the experiments indicate that the accuracy of a tree classifier rises to a certain limit as the number of clusters increases and subsequently decreases, emphasizing the importance of the proposed pruning criterion. Finally, the experimental results show that SVMs with linear kernel functions attain higher classification accuracy, which aligns with Cover's Theorem.

## 5. Conclusions

Fine-grained classification is a challenging problem in computer vision research. To address this issue, this paper proposes a self-supervised tree-structured framework that achieves accurate classification by mitigating the impact of the differences between clusters. The process of constructing a tree-structured classifier involves the following steps: (1) automatically constructing an initial classification tree using a clustering algorithm that groups similar subclasses and (2) building the tree-structured classifier by applying the proposed pruning criteria to eliminate unnecessary clusters from the tree hierarchy. By evaluating the classification accuracy and training time of the tree-structured classifier on diverse structures and datasets, this study demonstrates the robustness of the proposed method. The experimental results indicate that the proposed tree-structured classifier significantly enhances fine-grained classification accuracy and reduces training time relative to traditional CNN models.

**Author Contributions:** Conceptualization, Q.C. and L.N.; methodology, Q.C.; software, Q.C.; validation, Q.C., X.S., and H.D.; formal analysis, Q.C.; resources, L.N.; data curation, Q.C.; writing—original draft preparation, Q.C.; writing—review and editing, L.N.; visualization, Q.C.; supervision, L.N.; project administration, L.N.; funding acquisition, L.N. All authors have read and agreed to the published version of the manuscript.

**Funding:** This research was partially supported by the National Natural Science Foundation of China (No. 62006090) and Research Funds of Central China Normal University (CCNU) under Grants 31101222211 and 31101222212.

**Institutional Review Board Statement:** Not applicable.

**Informed Consent Statement:** Not applicable.

**Data Availability Statement:** This paper uses publicly available datasets, CUB-200-2011 is available at https://www.vision.caltech.edu/datasets/cub_200_2011/ (accessed on 27 February 2023); FGVC Aircraft is available at https://www.kaggle.com/datasets/seryouxblaster764/fgvc-aircraft; Stanford car is available at http://ai.stanford.edu/~jkrause/cars/car_dataset (accessed on 27 February 2023).

**Conflicts of Interest:** The authors declare that they have no known competing financial interests or personal relationships that could have appeared to influence the work reported in this paper.

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
