# Peer review of "A Self-Supervised Tree-Structured Framework for Fine-Grained Classification"

_applsci, doi:10.3390/app13074453_

Round 1

Reviewer 1 Report

The paper is very well written, but the inferences from the LS are fewer, so better to include more related reviews here. Also, the model testing or training is overfitting or not? How do you prove it? (Like 70-30/ 60-40 ratio).

Does the abstract summarize the paper's objectives, main thrust, and significant conclusions? Please consider whether or not the Abstract conveys the purpose of the study, provides a balanced and accurate depiction of the key findings, and addresses the implications of the work for Information Science. Could a person read the abstract and clearly understand what the article will be about? Will the keywords enable other professionals to locate the work with the search engines commonly used by academic libraries? What about the conclusion? Does the manuscript give a sense of revisiting the main ideas briefly? Does it make the reader feel that all the ideas have been tied together?

Similar types of work are already available: https://arxiv.org/abs/2107.13973v1

https://paperswithcode.com/task/fine-grained-visual-categorization

https://www.sciencedirect.com/science/article/pii/S0031320318304230

Can you list out the novelty of the work in a better way?

Author Response

Dear Reviewer,

Thank you very much for your insightful comments on our manuscript. We greatly appreciate your time and effort in reviewing our work.

We have carefully considered all your comments and have made revisions accordingly, and attached the file "response to reviewer 1". Firstly, we have included a response that lists your suggested comments and our corresponding responses. Following the response is the manuscript containing the revised parts according to your comments, highlighted for your convenience. Finally, we have attached the "English Editing Certificate" to provide proof that our paper has undergone language editing services provided by Author Services at the end of the file.

We hope that the revised manuscript addresses your concerns and meets your expectations. Thank you again for your valuable feedback, which has greatly improved the quality of our work.

Sincerely,

Qihang Cai

Reviewer 2 Report

The authors of the paper entitled “A Self-Supervised Tree-Structured Framework for Fine-Grained Classification” proposed a framework for fine-grained classification which is a challenging problem in computer vision field. The paper compares the framework with traditional convolutional neural network (CNN) strategy and states its results. The overall structure of the paper is good however, the paper needs a thorough proofread for typos and grammatical issues e.g. Line 80: correct use of term “speech recognition” instead of “speech recognization”, Figure 8 caption, Line 325, Line 360, Line 136: Duplicate words “following”. Moreover, the experimental section needs major revision and the paper should be revised as per the following comments.

 The paper does not define the term “machine-learning matcher” that authors have used in the  study. The abstract section does not state the improvement achieved by the proposed approach over traditional CNN.

The related work section must be more elaborated.

Section 3 provides a comprehensive detail in a good manner of the proposed work. However, it is suggested to start the section by stating the framework rather than discussing the general information.

The experiment section needs thorough revision. The details of experiments should be clearly presented along with the results. The section misses the introductory part as well as Section 4.1.1 “Conducted Experiments” describes experiments in an ambiguous manner. It is suggested to revise the section heading and provide a list of experiments that are being performed.

The experimental setup is not clearly defined and the parameter settings for the proposed work are missing. The discussion of the experimental results is also confusing and does not clearly mention the version of the classification model being used. Table 2 and Table 3 should be revised. What is actually addressed by the term “Tested Matcher” in Table 4; the term is not clearly defined before and the distinction between traditional and proposed approach is invisible in the table.  

Justification of the proposed approach is not satisfactory in terms of results.

Why only accuracy is used for performance evaluation? It is recommended to evaluate the proposed work using other metrices.  

Author Response

Dear Reviewer,

Thank you very much for your insightful comments on our manuscript. We greatly appreciate your time and effort in reviewing our work.

We have carefully considered all your comments and have made revisions accordingly, and attached the file "response to reviewer 2". Firstly, we have included a response that lists your suggested comments and our corresponding responses. Following the response is the manuscript containing the revised parts according to your comments, highlighted for your convenience. Finally, we have attached the "English Editing Certificate" to provide proof that our paper has undergone language editing services provided by Author Services at the end of the file.

We hope that the revised manuscript addresses your concerns and meets your expectations. Thank you again for your valuable feedback, which has greatly improved the quality of our work.

Sincerely,

Qihang Cai

Reviewer 3 Report

The approach presented attempts to improve the object classification by applying a self-supervised hierarchical classification. The corresponsding tree construction is clearly presented and could be of benefit to the research community - when suitable application cases arise. 

A major justification why I prefer publication is in place is that this kind of aproach (besides refreshing and well-documented) could produce in some particular applications larger benefits than with the covered 3 test cases. 

References cover the field well. 

p. 5: clustrer --> cluster (editors could take care of this)

Author Response

Dear Reviewer,

Thank you very much for your insightful comments on our manuscript. We greatly appreciate your time and effort in reviewing our work.

We have carefully considered all your comments and have made revisions accordingly, and attached the file "response to reviewer 3". Firstly, we have included a response that lists your suggested comments and our corresponding responses. Following the response is the manuscript containing the revised parts according to your comments, highlighted for your convenience. Finally, we have attached the "English Editing Certificate" to provide proof that our paper has undergone language editing services provided by Author Services at the end of the file.

We hope that the revised manuscript addresses your concerns and meets your expectations. Thank you again for your valuable feedback, which has greatly improved the quality of our work.

Sincerely,

Qihang Cai

Round 2

Reviewer 2 Report

The paper titled “A Self-Supervised Tree-Structured Framework for Fine-Grained Classification” has been improved greatly on the recommendations; however following changes are still needed:

  • The description of Experiment 4 is ambiguous Lines 286-289
  • Line 337-338, Line 389-390. rephrase suggested to increase the understanding and readability
  • Values of the tested matcher in Table 6 are still unclear; what is exactly meant by 2-1, 5-1, etc. should be clearly defined
  • The traditional CNN model used in other experiments is ResNet, however, in Experiment 4 the definition of traditional CNN becomes confusing as ResNet is selected for the proposed work too
  • Please clear the term "LS" used in response to Comment 4, do you mean literature study?

Author Response

Dear reviewer,

Thank you for taking the time to review our manuscript and for providing such constructive comments. We appreciate the effort you have put into reviewing our work and helping us improve it.

We have revised the manuscript according to your constructive comments. Following the response is the manuscript containing the revised parts according to your comments, highlighted for your convenience.

We hope that our revisions have addressed your concerns. Thank you again for your time and effort in reviewing our manuscript.

Best regards,

Qihang Cai
